# Odor Annoyance Assessment by Using Logistic Regression on an Example of the Municipal Sector

**Alicja Wroniszewska [1,*] and Jerzy Zwoździak [2]**

[1]  Department of Environment Protection Engineering, Wroclaw University of Science and Technology, 50-370 Wroclaw, Poland

[2]  Faculty of Safety Science, General Tadeusz Kosciuszko Military University of Land Forces, 51-147 Wroclaw, Poland; jerzy.zwozdziak@awl.edu.pl

*   Correspondence: alicja.wroniszewska@pwr.edu.pl

**Abstract:** Around the sewage treatment plant, in the area affected by a large number of complaints about odor annoyance, field measurements of odor properties and questionnaires were carried out. It was confirmed that the inhabitants of the zone closest to the plant are most exposed to the smell, the most intense smell comes from the sludge dryer building, and smells from primary settling tanks and sediment plots are perceived as unpleasant. The analysis of surveys confirmed the problem of odor nuisance, especially in the immediate vicinity, where over 50% of respondents considered odor annoyance as extreme. A division of respondents was introduced into those experiencing severe nuisance and those for whom the smell was not annoying. Then, to relate the probability of occurrence of odor nuisance with a group of independent variables, logistic regression was used to describe the impact of independent variables on the dichotomous dependent variable. It has been shown that the likelihood of experiencing odor nuisance increases with the increase in the intensity of current odors, the parallel noise, and in people who focus on the existing smell, and decreases with increasing satisfaction with their health and in the case of regularly occurring odor.

**Keywords:** odor annoyance; field measurements; logistic regression

## 1. Introduction

For several years, the number of complaints about the odor nuisance around plants from the municipal and household sector has been growing. In many cases, unfortunately, odorous gases come from facilities that are aimed at improving the natural environment and the quality of people's life. These are mainly municipal facilities, such as sewage treatment plants, waste disposal or storage plants, or biogas plants. Often, their poor location and unfavorable meteorological conditions, as well as the terrain contributing to the good spread of odors, mean that instead of improving the state of the environment and people's feelings, they further aggravate the impressions of residents living nearby.

Data on the state of the environment are today publicly available. This influences the growing interest of people in environmental protection. Public consultations are also becoming widespread, enabling citizens to feel their contribution to sustainable development. It also causes, along with the growth of economic development, an increasing number of complaints related to environmental pollution from year to year, also in terms of odor annoyance. The lack of uniform legal regulations regarding this issue in the European Union countries has additionally slowed down the introduction of the act on odor nuisance in Poland. In most cases, complaints about unpleasant odors are directed to Sanitary Inspectorates, which indicates that people identify the smell problem as threatening their health and life.

According to studies on the nuisance caused by, among others, smell [1–3], adverse reactions of people to the smell that appears in their environment is a very complex problem. It consists of many factors, not only related to the scent itself, but also the so-called nonolfactory factors that can strengthen or weaken these reactions, such as age or health problems, but also our commitment to environmental protection or inclination to social activity.

In order to assess the impact of a smell on the health and well-being of the inhabitants of areas adjacent to the source of odors, two types of research can be conducted: survey [4–7] and field research [7–13]. The aim of the questionnaires is to identify the reactions of people living in areas exposed to the presence of unwanted odors and to designate an area where the problem can be observed. On this basis, it can be assessed whether there is any odor nuisance in a given area. On the other hand, field studies are carried out to assess the properties of fragrances, i.e., their frequency, intensity and hedonic tone. Field measurements are designed to determine how much the inhabitants of the studied area are exposed to the presence of odors from nearby sources, how much the present smell is intense, and whether it is perceived as pleasant, unpleasant, or neutral. The field measurements, moreover, are carried out by a selected panel of experts who can assess the surrounding smell without negative associations. Each of these tests individually helps to assess the current state of a given area in terms of odorous air quality. However, none of them separately is sufficient for a comprehensive assessment. Therefore, this study combined both types of research.

In studies of odor nuisance, we often deal with a dependent variable of the dichotomous type [14]; for example, 1—we feel a given smell negatively, 0—we do not feel it; 1—headache occurs, 0—does not occur. Then, one can ask the question which of many considered independent variables (age, gender, distance from the plant, and others) significantly affects the occurrence of the symptom? In such situations, we can use logistic regression. In the construction of the model to assess odor nuisance, therefore, the combined results of field measurements (odor properties and surveys) carried out in and around a large sewage treatment plant were used, and logistic regression was used to interpret the data.

Therefore, the main goal of the work was to build a model for assessing the impact of external factors on the perception of odor nuisance based on logistic regression. The scope of work included:

- Estimation of the frequency, intensity, and hedonic tone of odors occurring in the studied area;
- Assessment of perceived odor annoyance by residents living in areas adjacent to the source of odors.

## 2. Materials and Methods

The investigations were carried out around the sewage treatment plant, which is located in the Wielkopolska Voivodship, in rural areas, at the city border. The plant is located west of the residential buildings at a distance of about 200 m. The housing estates in the village adjacent to the sewage treatment plant are mostly situated in hilly areas. West and southwest winds prevail.

It is a mechanical–biological sewage treatment plant with increased biogenic removal and full treatment of sewage sludge. The facilities of this sewage treatment plant enable the reception of wastewater in an average amount of 200,000 $m^3$/day (in the rainy season up to 260,000 $m^3$/day). The treated sewage receiver is the river located west of the plant. The sewage treatment plant is charged with PE = approx. 1,000,000.

According to the field research methodology used in European countries (VDI 3940) [15], the first stage of the measurements was to determine the area of research (Figure 1).

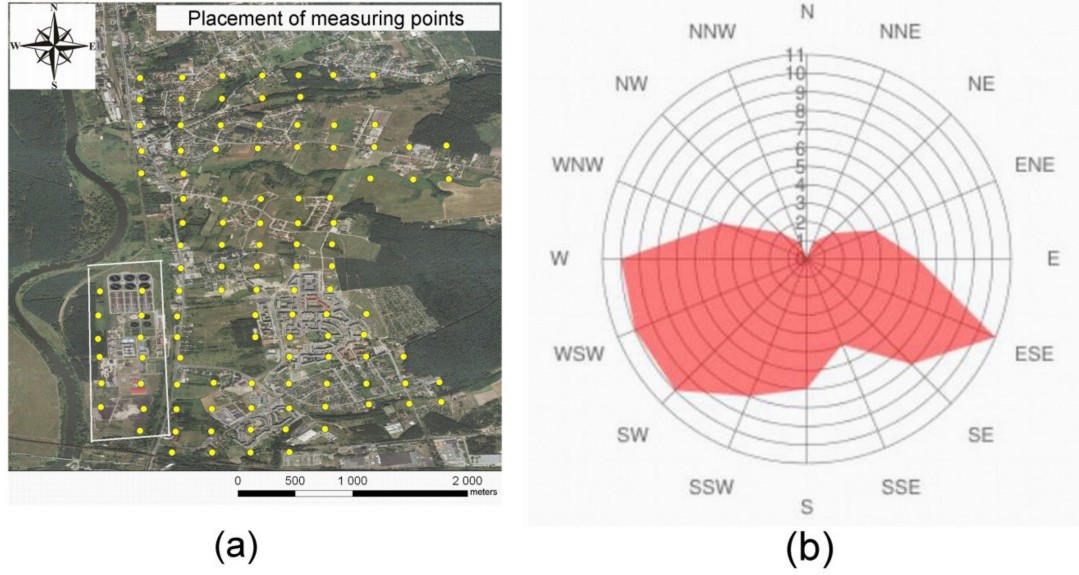

**Figure 1.** (**a**) Measuring grid designated for field measurements of odor properties and assessment of annoyance with the plant marked (based on [16]); (**b**) annual wind rose for the nearest city.

For this purpose, the dispersion of odor was calculated first [17]. Odor concentration calculated using the reference model at receptor points initially indicated the possibility of occurrence of odor concentration levels above 1 $ou_E/m^3$ in the analyzed area. However, due to model assumptions that do not include, among others, changes in weather conditions, terrain, or variable emission of pollutants, the model turned out to be not very reliable at determining the area exposed to the effects of odors in the plant. According to the results of the model, it should cover areas where the smell should no longer be felt due to the terrain and naturally occurring barriers (wide green belt, river, and floodplains). It was, therefore, necessary to carry out field reconnaissance and thorough examination of the area potentially exposed to the smell emission. The results of the inspection showed that the reference model incorrectly determined the area potentially exposed to odors. Therefore, eventually, the research area covered the area marked in Figure 1 (p. 3).

On this basis, the boundaries of the "measuring area", i.e., the measuring grid, were indicated, which was then divided into 94 "measuring squares". Each measuring square consists of "measuring points" (104), in which the frequency, intensity, and hedonic tone were measured. The entire research area was then divided into three zones. Zone 1 was located near the plant, zone 2 was about 500 m to the east, and zone 3 was furthest from the plant (about 1 km to the northeast).

During the entire study period (summer and autumn–winter season 2011), 520 individual measurements were carried out, based on which the frequency of the most characteristic odors, their intensity, and hedonic tone were assessed.

A 16-person, adequately, according to [18], trained group of testers took part in the field measurements. To guarantee the highest reliability of measurements, during the entire study period, 5 measurements were made at each point at different times of the day (morning, daytime, evening, night), by various evaluators, and the points were randomly assigned to an expert and measurement day and time. The time of a single measurement was about 10 min. At that time, each person (a member of the expert group), facing the wind, breathed in the surrounding air every 10 s, and marked on the protocol whether they sensed any odor and gave its type (e.g., preliminary settling tanks, sludge dryer) using the appropriate code. Then, based on measurements, it was possible to determine the frequency of occurrence of a given type of fragrance in the so-called odor hours. One "odor hour" meant that the smell was felt over 10% of the time of one measurement, i.e., over 6 times. The incidence of odor in a given measuring square was calculated as the arithmetic mean of the frequency at the measuring points constituting it.

The second stage of field research, after measuring the frequency of smell, was the study of its intensity, i.e., the strength with which the odor is felt. The person conducting the measurements assessed only the intensity of odors characteristic of the source being tested. The maximum intensity of the strongest impression of a given type of fragrance received during one measurement cycle (10 min) was indicated, the frequency with which this impression occurred was recorded, and at the end of the measurement average intensity impression was noted.

To assess the intensity of the smell in the field, the intensity scale from 1 to 6 was used. It was assumed that the value "6—extremely strong" corresponds to the intensity of the smell felt as stronger than "5—very strong", and the lowest value on the scale, "1—very weak", meaning exceeding the odor recognition threshold. The average value, "3—distinct", did not only indicate the recognition of a given fragrance but indicated that it was stronger than "2—weak", although weaker than "4—strong".

During calculating the average intensity, it was essential to consider the sensitivity of smell in the field. For this purpose, the Weber–Fechner factor $F_w$ was used. Based on earlier studies [19], in which coefficients between 1.5 and 2.5 were tested, it was found that the coefficient 2 was best matched because between the odor emission and the perceived annoyance, the strongest relationship occurred for this value.

The final stage of the study was the assessment of the odor hedonic tone, i.e., the effect caused by the odorant, classified between "extremely pleasant" and "extremely unpleasant" (VDI 3882) [19,20]. In this case, only source-specific odors were considered. The evaluation consisted of 3 parts. In the first, the evaluator marked on the protocol the hedonic tone of a given fragrance, which was the most pleasant in a given measurement cycle. In the second part of the assessment, the hedonic tone was highlighted for the least pleasant smell. Finally, the average hedonic quality value was indicated. To assess hedonic tone, a scale from −4 to 4 was used. In contrast to the odor intensity scale, points on the hedonic tone scale were sequenced but had a center and two divergent ends. The average hedonic tone at a given measuring point was calculated by converting values from −4 to 4 for positive values from 0 to 9 [20,21].

During each measurement, some meteorological data were recorded, such as wind direction (using a compass), cloud cover, or possible rainfall.

Surveys were conducted based on German guidelines of the VDI 3883 series [22]. During several days, a group of interviewers visited residents of housing estates included in the measurement area and conducted interviews with them for about 15 min, asking questions and marking answers on the forms. The respondents were chosen randomly and were only adults. The survey contained questions about demographic data, as well as the state of the environment and its pollution, with an emphasis on air pollution by smell and nuisance caused by it. Table 1 summarizes the data on the number of interviews conducted in each zone, as well as the number of residents who could not be contacted or who refused to respond. The basic statistical data of the respondents are presented in Table 2.

**Table 1.** Summary of the number of surveys carried out and not carried out.

| Questionnaires | | | |
|---|---|---|---|
| **Zone 1** | **Zone 2** | **Zone 3** | **Sum** |
| **Done** | | | |
| 20 | 40 | 48 | 108 |
| **Refusal** | | | |
| 2 | 46 | 14 | 62 |
| **Absent** | | | |
| 22 | 96 | 49 | 167 |

**Table 2.** Basic statistical data on respondents.

| Age | | Gender | | Education | |
|---|---|---|---|---|---|
| **19–25** | 3 | **Male** | 43 | **Basic** | 2 |
| **26–40** | 39 | | | **Secondary** | 24 |
| **41–60** | 37 | **Female** | 57 | **Vocational** | 29 |
| **>60** | 21 | | | **High** | 35 |

An essential part of the survey were questions about the perceived odor annoyance on an 11-point numerical scale (from 0 to 10) and a descriptive scale (from "not at all" to "extremely annoying"). Additionally, the comparison of these two scales allowed us to assess the reliability of the answers.

Since odor nuisance may depend, to a large extent, not only on the intensity and frequency of the smell but also other factors that are not olfactory, such as age, health, or degree of attachment to the place of residence, the survey also included questions about these factors (e.g., period of residence in a given location, use of a balcony or garden, health problems, and accompanying environmental pollution, as well as personal data such as age, gender, education). In order for the respondents not to focus only on the fragrance, the survey was introduced in parallel to the questions about the perceived odor nuisance, with questions about the perceived nuisance caused by other pollution, in this case, noise.

Questionnaires took place in the same area where the field measurements were carried out. Each surveyed person was assigned a measurement square number, thanks to which, using statistical analysis, it was possible to compare later the frequency, intensity, or hedonic tone of the smell felt by a group of experts with the odor nuisance felt by residents.

Statistical analysis of both types of measurement data was carried out using the Statistica 10 software. The purpose of the analysis was to find a link between the probability of occurrence of odor nuisance and a group of independent variables, such as age, gender, education, coexisting environmental pollution, subjective health, type of character (problem-oriented), amount of time spent at home, and the smell present in the area: frequency, intensity, and hedonic tone. Logistic regression was used in this analysis. It is a mathematical model that can be used to describe the impact of independent variables on a dichotomous dependent variable. Let $Y$ denote a dependent variable with values: 0—will not feel odor annoyance ($0 \leq OA \leq 6$), 1—will experience odor annoyance ($7 \leq OA \leq 10$) [23]. The logistic regression model for such a dichotomous variable took the form:

$$P(Y = 1 | x_1, x_2, \ldots, x_k) = \frac{e^{(a_0 + \sum_{i=1}^{k} a_i x_i)}}{1 + e^{(a_0 + \sum_{i=1}^{k} a_i x_i)}} \tag{1}$$

$a_i, i = 0, \ldots, k$—regression coefficients; $x_1, x_2, \ldots, x_k$—independent variables.

The backward step-wise method was used; however, the use of other available estimation methods did not cause any visible changes in the form of the model and the values of the regression coefficients. The ratio of the product of correctly classified cases to the product of incorrectly classified cases reached a value in excess of unity, which means a classification much better than that which can be expected by accident. The level of test probability for the Homer–Lemeshow test $p = 0.1134$ and the value of $R_N^2 = 0.53$ indicated that there were no grounds for rejecting the hypothesis about a good fit of the model to the data. The odds ratios for the individual change of the analyzed parameters were also calculated.

The measure of good fit of the model, which determines its good predictive properties, is the ROC (receiver operating characteristic) curve and measures based on the classification matrix [24]. The logistic regression model returns, inter alia, the probability value of belonging of individual cases to the modelled class. Often the values of this probability are used to make a decision. For this

purpose, a certain point in the probability value is taken as the cut-off point. Above this point we assume the occurrence of the modelled class, below the opposite event. Since the model is only an approximation of real processes, the adopted division is likely to result in incorrect assignments to both analyzed classes. As a result of the adopted cut-off point, the analyzed cases can be assigned to the following states: correct indication of the distinguished class (TP—true positive) and correct failure to indicate the other class (TN—true negative) in a situation where no errors were made. On the other hand, errors are made when the distinguished class is incorrectly indicated (FP—false positive) or the distinguished class is not indicated when it should be indicated (FN—false negative).

In Table 3, TP, FP, FN, and TN represent the number of observations that hit a given cell in the table. It summarizes the classification results for a given decision rule, comparing the actual state with the indication of the model.

**Table 3.** Classification matrix—the actual state and indication of the model.

|  | **Highlighted State Observed** | **The Highlighted State Was Not Observed** |
|---|---|---|
| The distinguished state was expected | TP | FP |
| The distinguished state was not expected | FN | TN |

A good model is one that minimizes the number of errors, i.e., FN and FP. The best decision rule is to ensure the best results—that is, the minimum number of errors. Thus, measures of the quality of decision rules are introduced and two main measures are defined: specificity and sensitivity.

Sensitivity is defined as:

$$\text{Sensitivity} = \frac{\text{TP}}{\text{TP} + \text{FN}}. \tag{2}$$

In other words, sensitivity indicates the fraction of objects that actually belong to the state distinguished by the model that have been correctly classified.

Specificity is defined as:

$$\text{Specificity} = \frac{\text{TN}}{\text{TN} + \text{FP}} \tag{3}$$

Specificity indicates the fraction of objects belonging to the state not distinguished by the model that have been correctly classified. Both sensitivity and specificity are the basis for constructing the ROC curve.

The classification matrix allows us to evaluate the quality of the decision rule for a specific cut-off point of the model response. In order to assess the predictive ability of the model, regardless of the adopted cut-off point, the ROC curve is used.

## 3. Results

### 3.1. Field Measurements

West and southwest winds prevailed during the measurements (Figure 2). Due to this fact, areas located to the east and northeast of the plant, i.e., residential estates, could be the most exposed to the impact of odor emissions.

The results presented in Figure 3 (p. 7) confirm that the most vulnerable to the presence of odor were the inhabitants of zone 1, closest to the sewage treatment plant, about 200 m east of the plant border, marked in the figure as a rectangle. It can be seen that the fragrance moved according to the wind direction, and the frequency decreased with distance.

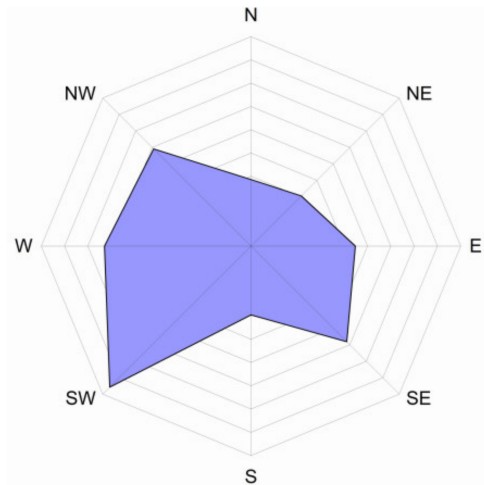

**Figure 2.** The wind rose based on measurement results.

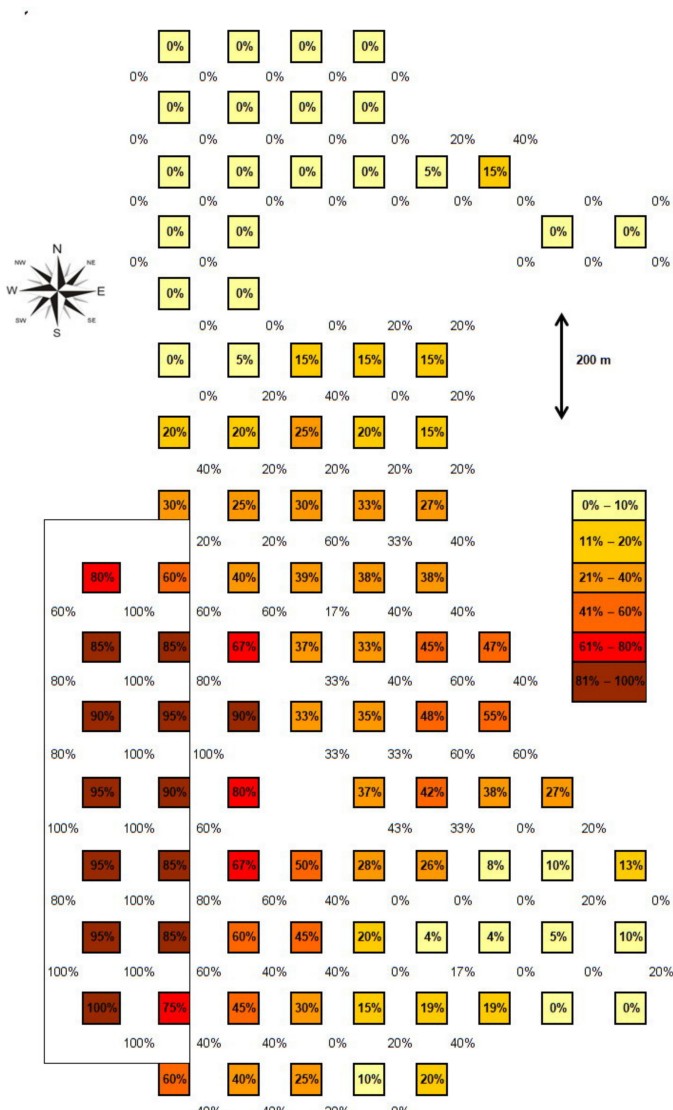

**Figure 3.** The odor frequency of the sum of all odors originating from the plant in the study area (based on [16]).

The results of the average intensity of individual types of fragrances are summarized in Table 4. The results of odor intensity measurements show that the most intense odor came from the sludge dryer building (4—strong), and the average intensity of the remaining odors throughout the entire study area was 3—distinct.

**Table 4.** Average intensity (on 1–6 scale) of odors from the analyzed plant.

|  | SETTLERS | SEDIMENT PLOTS | SLUDGE DRYER | BIOREACTORS |
|---|---|---|---|---|
| **Average intensity** | 3 | 3 | 4 | 3 |
| **Average hedonic tone** | 4 | 4 | 3.5 | 6 |

The results of measurements of the odor hedonic tone from selected sources recorded throughout the field research show that while odors from bioreactors were received by most assessors as somehow pleasant or neutral (average hedonic tone 6), the remaining odors were, in most cases, assessed as definitely unpleasant: from preliminary settling tanks and sediment plots—4, and sludge dryer—3.5. Reception of odors from sewage treatment plants as pleasant could only be caused by the periodic stay of assessors in the studied area.

### 3.2. Questionnaires

After analyzing the results of surveys, it can be concluded that the described sewage treatment plant was the main source of the smell in the studied area. This was indicated by the vast majority of respondents (over 80%). Other answers to the question about the source of the fragrance were "other" (3%), including the smell of exhaust gases or smoke from domestic boilers and the "poultry plant" (8%) located in the study area. However, the last one was not identified during field tests. This may be because this type of smell was perceptible in earlier years, and the respondents' answers are often distorted just by feelings from the past. In total, 7% of respondents could not indicate any source (Figure 4a). To the question: "Please describe the most disruptive smell felt in this area", the vast majority of respondents (over 80%) answered that they associate the smell present in this area with a sewage treatment plant. Fragrances described as feces (over 36%), rotten eggs (almost 28%), or chemical (over 16%) were typically the most characteristic smells coming from the plant: primary settling tanks, bioreactors, and sediments (Figure 4b).

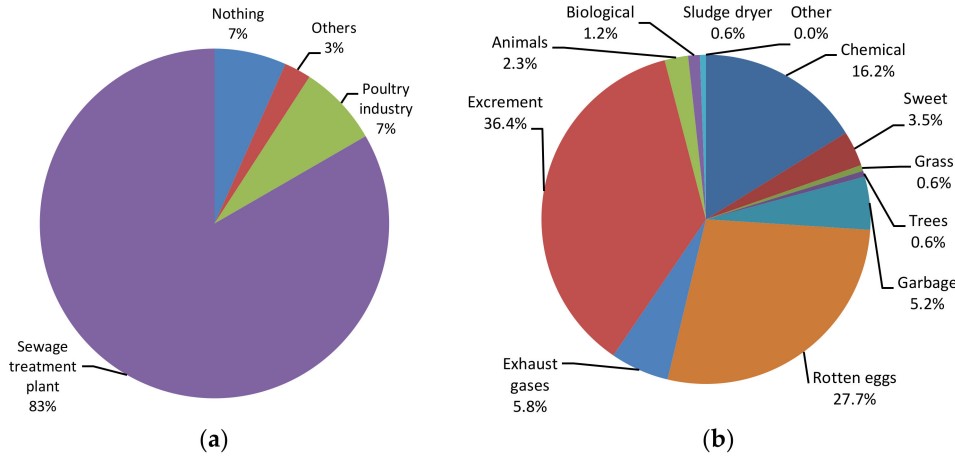

**Figure 4.** Share of answers to the question about the source (**a**) and the type (**b**) of smell.

Figures 5 and 6 summarize the survey results regarding the perceived annoyance by residents on two scales: an 11-point numerical scale and a descriptive scale.

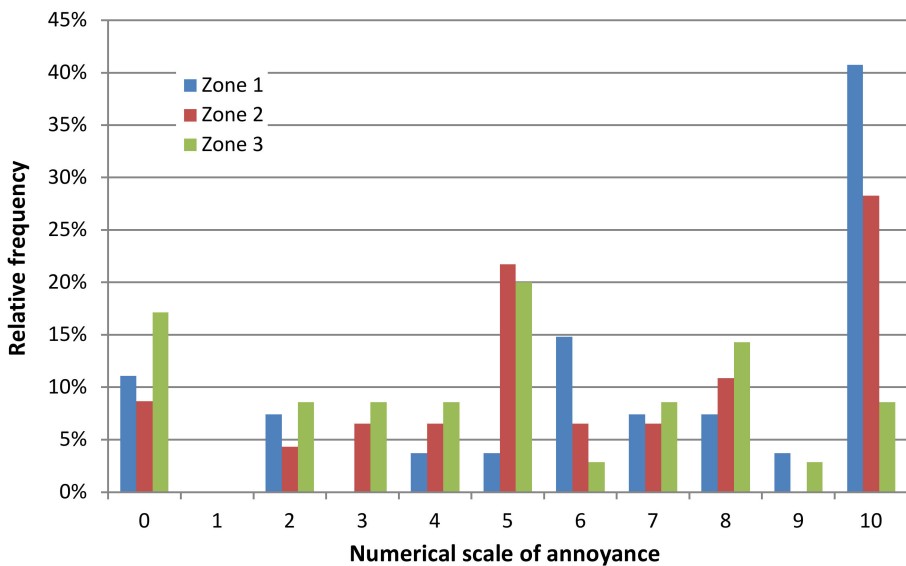

**Figure 5.** Comparison of odor annoyance expressed on a numerical scale felt in the examined area in 3 zones (based on [16]).

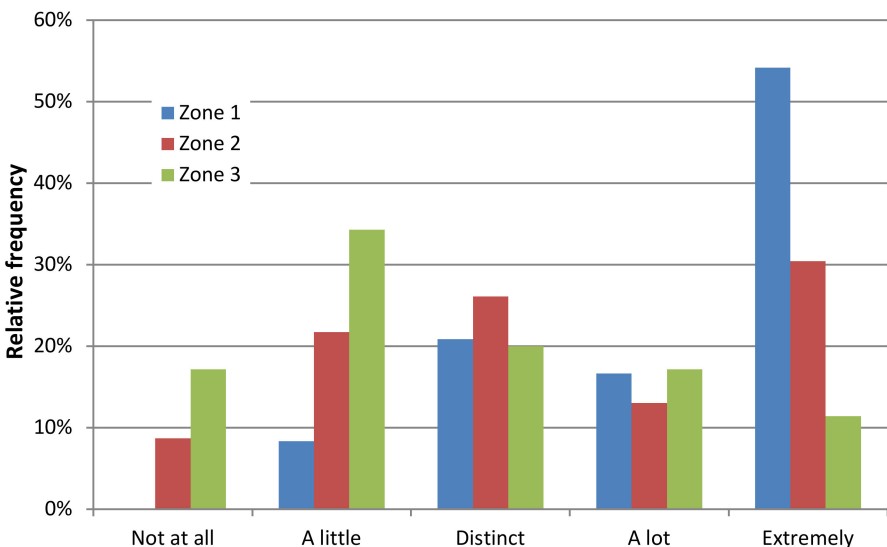

**Figure 6.** Comparison of odor annoyance expressed on a descriptive scale felt in the examined area in 3 zones (based on [16]).

Survey results confirmed the problem of odor nuisance in the study area, especially in zone 1, where over 50% of respondents believed that odor nuisance was extreme ("extreme" odors on a descriptive scale). Also, the results from the 11-point numerical scale (0–10) indicated the biggest problem in zone 1, where only 20% of respondents believed that odor nuisance was below 5, and almost 60% indicated nuisance of at least 7. In other zones, the problem also existed, although the answers were less clear-cut. Over 45% and about 30% of respondents from zones 2 and 3, respectively, believed that the odor nuisance was at least 7 and was "very" or "extremely" annoying.

Figure 7 presents the answers to the question: "Please indicate which of the following health complaints have occurred in the last two years or still occur. Please indicate only the ailments, the causes of which have not been diagnosed by physicians", for the whole surveyed population. Over 30% of people complained of headaches, more than 20 of breathing problems, and more than 10% of unexplained nausea.

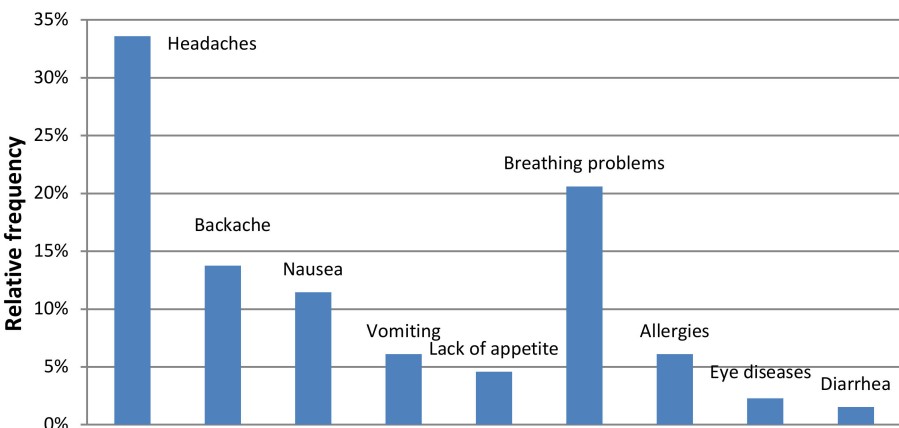

**Figure 7.** Answers to the question about the perceived health ailments not diagnosed by the doctor in the entire study area.

Figure 8 presents the respondents' answers to the question: "How do you assess your health generally?" Although the vast majority of respondents described their health as "good", it can be seen that more than 40% of people in Zone 1 described it as "bad". It is also clearly seen that in Zone 3, health was felt best.

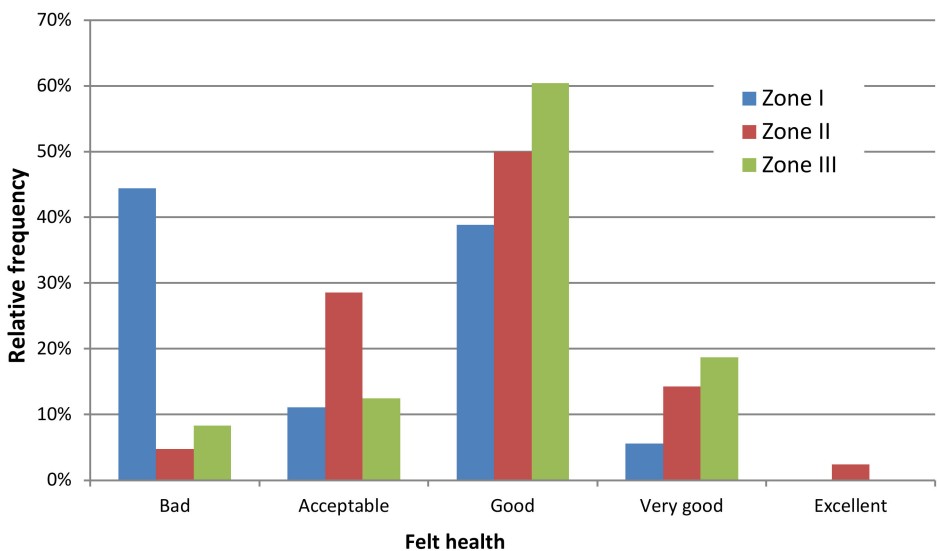

**Figure 8.** Comparison of answers to the question about the state of health in 3 examined zones.

Figure 9 summarizes the answers to the question: "Please indicate how much each of the following factors negatively affecting the environment bothers you." It can be stated that the biggest problem concerned air pollution (smoke, odors, dust) and noise (cars, factories, neighbors).

Figure 10 presents the share of respondents' answers to the question: "What do you think or do when smells become annoying? Given the last 12 months while at home, how often have you thought or done the following?" The division into "never" and "very often" can be seen. Over 30% of respondents often or very often are typically very active (they complain or wonder how to solve this problem).

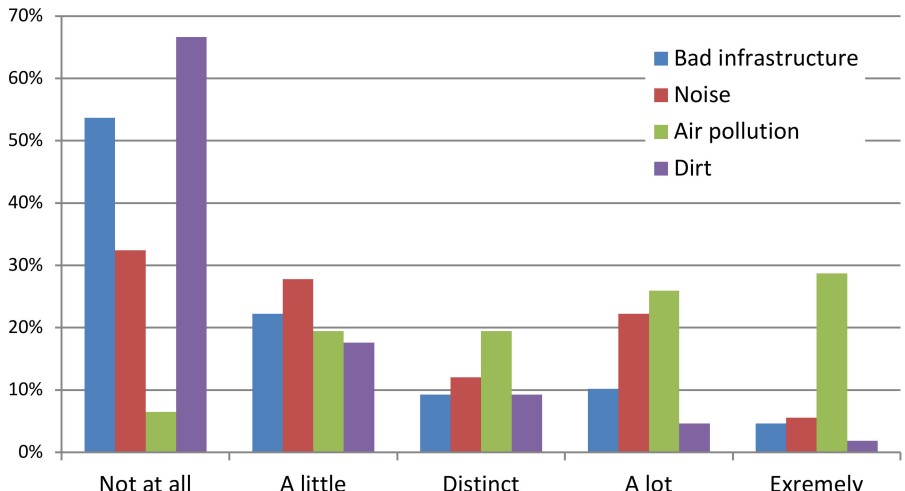

**Figure 9.** Questions about problems and general environmental pollution in the study area.

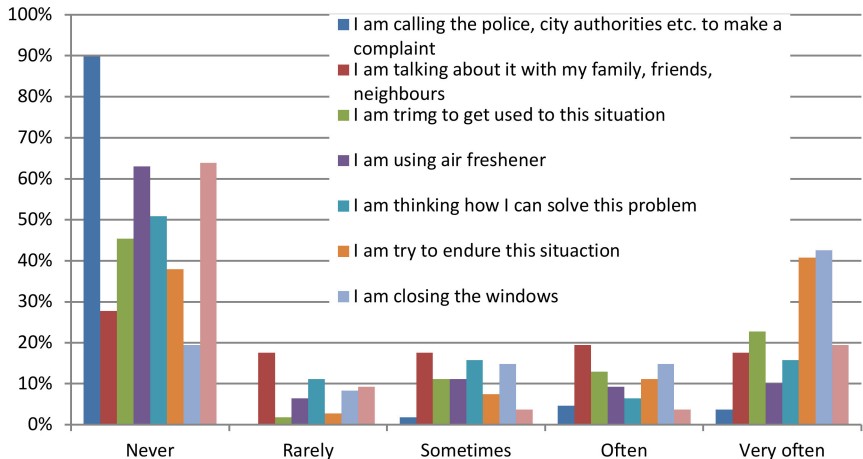

**Figure 10.** Questions about problems and general environmental pollution in the study area.

Figure 11 shows the spatial distribution of respondents' answers to the question about the perceived odor nuisance in the studied area. A division has been introduced between answers from 0 to 6 for those who did not experience significant nuisance (yellow), and from 7 to 10 for those who described the perceived nuisance as very serious (red). The arrangement of individual responses raises some doubts as to the source of the perceived annoyance. Both in the near and distant surroundings of the plant people experienced a serious and minor nuisance.

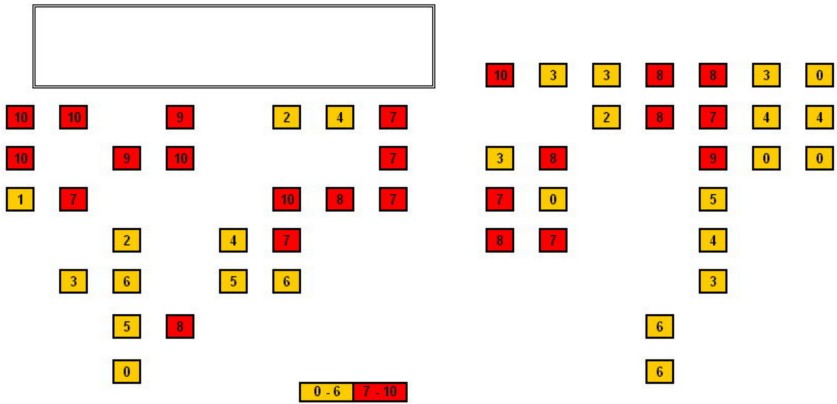

**Figure 11.** Perceived odor annoyance in measuring squares.

Summing up the results of this part of the research, it can be stated that odors in the studied area may affect housing estates located close to the plant. Survey results indicate that other parallel pollution, such as noise, also occurs in this area, which may increase the feeling of the nuisance.

### 3.3. Logistic Regression

The presented measurement results have shown that there is an odor nuisance in the studied area and that the smell—its frequency, intensity, and hedonic tone—can potentially have a negative impact on inhabited areas. However, there are inconsistencies (Figure 12). Many regions (squares), where there was an odor annoyance, coincided with the areas where there was an odor (red color), and there are also squares where neither odor nuisance nor odor were found (white color). However, there are many areas where there was a smell, but the residents did not complain about the nuisance, as well as such where the smell has not been registered by the assessors, and there was a severe nuisance caused by it.

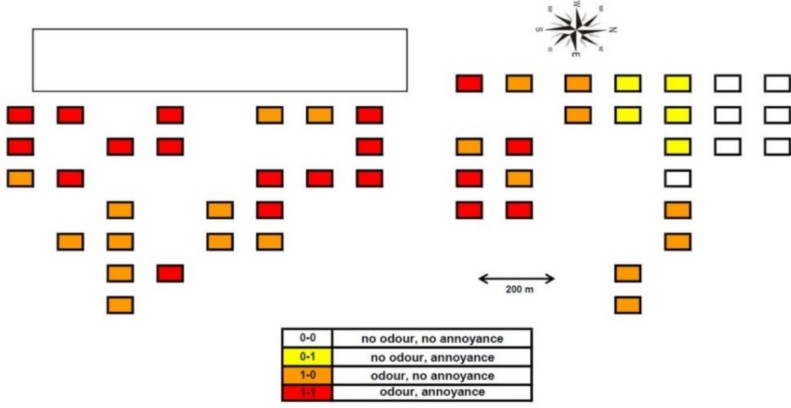

**Figure 12.** The occurrence of odor and perceived odor annoyance.

This may be because field studies are only carried out for a limited period, to a limited extent, while residents receive odors regularly, which was confirmed in other studies [25]. The perception of smells can, therefore, be different. As mentioned earlier, the presence of other, not olfactory, factors may also influence this. The answer to the question: "What causes nuisance and to what extent?" is therefore fundamental.

Respondents' responses regarding the perceived odor nuisance indicated on a numerical scale (the so-called odor annoyance thermometer—OA dependent variable) suggest that in a given area, the smell may cause odor nuisance in residents, and thus an increasing number of complaints. As already mentioned, the scale of odor nuisance was an 11-point scale, from 0 to 10. It is therefore difficult to determine whether the respondent feels annoyance or not if he indicates on the scale, e.g., 5. According to the latest research [26,27], to facilitate this assessment, the respondents were divided into those who experienced severe nuisance and those for whom the smell was not annoying or not very annoying. As mentioned, it was assumed that answers from 0 to 6, inclusive, meant that the respondent did not feel any severe nuisance, and from 7 to 10 that the perceived nuisance was significant. Each of the respondents was assigned a dichotomous type variable. Therefore, in further analyses, the situation in which the variable OA takes the value "0" will mean that the person did not feel the nuisance associated with the occurrence of smell, while the value of "1" indicates the person who experienced severe odor nuisance.

The logistic model, based on the results of field measurements of odor properties, as well as surveys and selected nonolfactory factors, is discussed below. Estimations of the model parameters were carried out for the whole examined population (N = 108). The presented logistic model contains only those independent variables which, as a result of the analysis, proved to be statistically significant (at $p < 0.05$).

The values of the model parameter estimators were statistically significant ($p < 0.05$). Based on the LR (likelihood ratio) test, it was also found that the addition of SEDIMENT and SETTLERS variables to the model did not significantly improve the model ($p = 0.15$, $0.16$). Still, the addition of the DRYER, ACTION, FREQUENCY, NOISE, and HEALTH proved to be significant ($p < 0.0001$).

The model shows (Table 5) that the increase in the likelihood of experiencing odor nuisance increases with increasing values of variables: "ACTION" (OR = 2.113), "DRYER" (OR = 2.195), "SEDIMENT" (OR = 1.425), "SETTLERS" (OR = 2.325), "NOISE" (OR = 1.405), and decreases with increasing values of the variables: "FREQUENCY" (OR = 0.001) and "HEALTH" (OR = 0.520).

**Table 5.** Selected results of the estimation of logistic regression model parameters for the examined population.

| | VARIABLE | | | | | | |
|---|---|---|---|---|---|---|---|
| | **ACTION** | **DRYER** | **SEDIMENT** | **SETTLERS** | **FREQUENCY** | **NOISE** | **HEALTH** |
| **Estimated parameter value** | 0.748 | 0.786 | 0.354 | 0.844 | −7.397 | 0.340 | −0.655 |
| **Significance level** | | | | <0.05 | | | |
| **95% confidence interval for parameters** | 0.411 ÷ 1.085 | 0.350 ÷ 1.222 | 0.064 ÷ 0.644 | 0.523 ÷ 1.164 | −10.457 ÷ 4.338 | 0.235 ÷ 0.446 | −1.035 ÷ −0.274 |
| **Odds ratio for individual change parameter** | 2.113 | 2.195 | 1.425 | 2.325 | 0.001 | 1.405 | 0.520 |
| **95% confidence interval for odds ratio** | 1.509 ÷ 2.958 | 1.420 ÷ 3.394 | 1.066 ÷ 1.905 | 1.688 ÷ 3.202 | 0.000 ÷ 0.013 | 1.264 ÷ 1.561 | 0.355 ÷ 0.760 |

Figure 13 shows plots of changes in sensitivity and specificity for the different values of the cut-off of the model predictions.

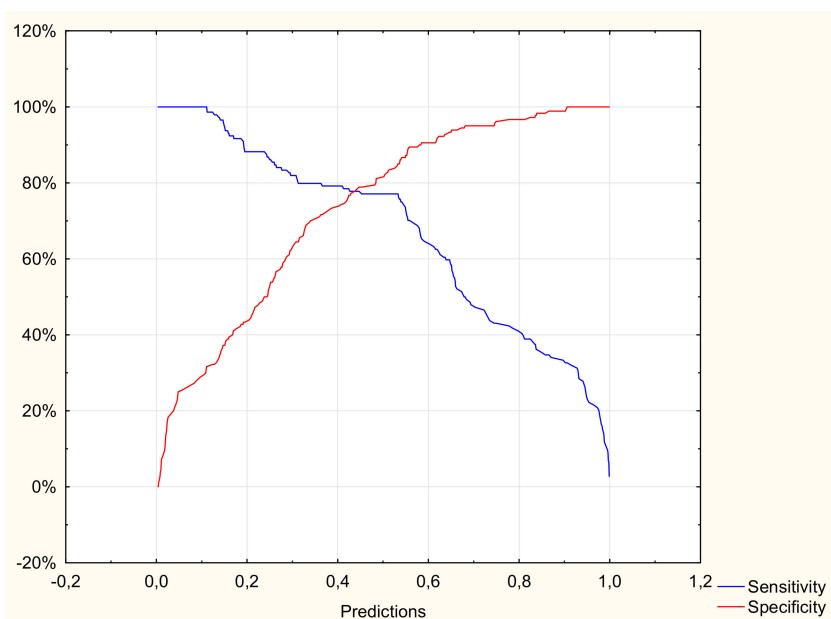

**Figure 13.** Graph of changes in sensitivity and specificity for different values of the cut-off points of the model predictions.

A good rule decision is one that maximizes both sensitivity and specificity. The sensitivity and specificity plots are then the basis for the creation of the ROC plot, i.e., the sensitivity versus specificity

(1—specificity) relationship for the obtained model for all possible cut-off points. If the ROC curve (Figure 14) is smoother, the more different the values of the indicator under study are.

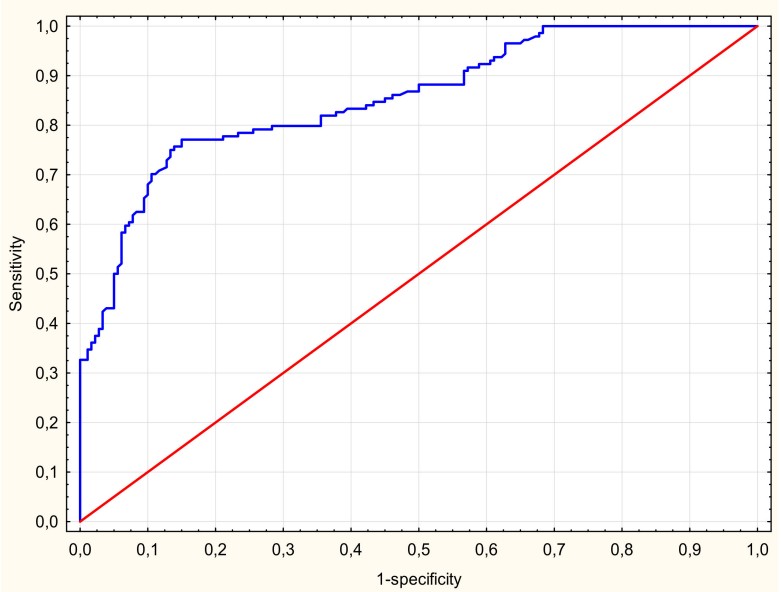

**Figure 14.** Receiver operating characteristic (ROC) curve.

The obtained ROC curve can now be used to evaluate the resulting model. For this purpose, the area under the plot of the ROC curve is calculated, denoted as AUC (area under curve). It is also a measure of goodness and validity of a given model. The AUC index is in the range [0,1]. The higher the AUC the better the model.

The value of the AUC field obtained in the tests was 0.82, which confirms a good fit of the model, and may also indicate its good predictive properties.

## 4. Discussion

The obtained results show that the probability of experiencing odor nuisance increases when the intensity of odors increases, together with parallel pollution, which is noise in this case (Figure 15), as indicated in a previous study [27]. This happens also for people of the problem-oriented type (Figure 16), which focus on the problem around them, such as annoying odors. Similarly, this probability decreases with increasing satisfaction with one's health (Figure 17), as proved before [28] and with regularly occurring odors.

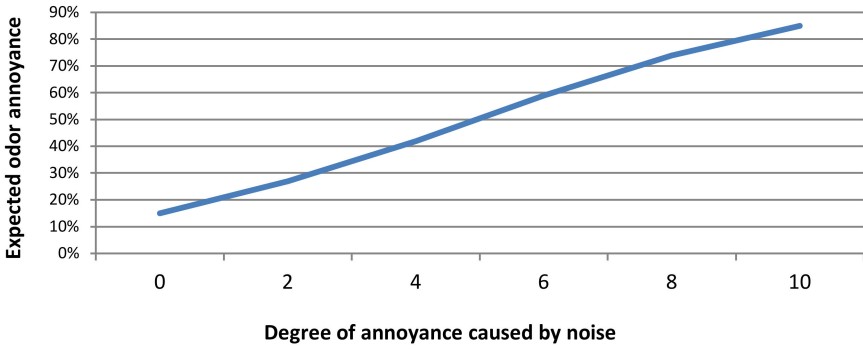

**Figure 15.** Expected odor annoyance depending on the degree of nuisance caused by noise.

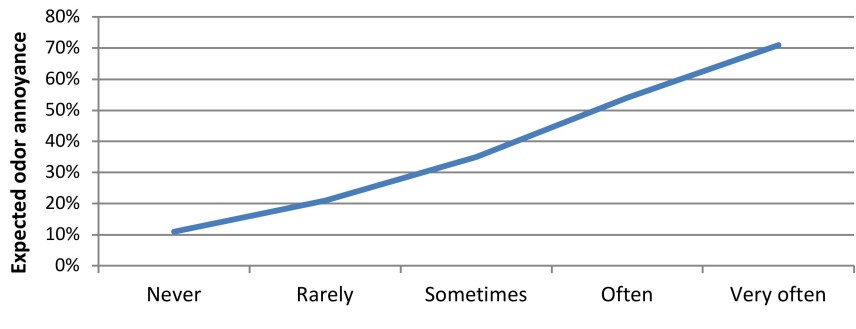

**Figure 16.** Expected odor annoyance depending on the frequency of taking action in stressful situations.

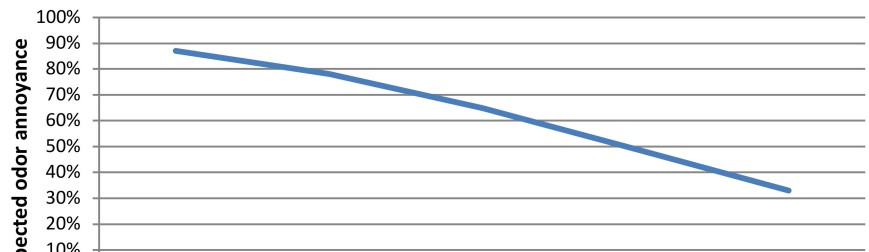

**Figure 17.** Expected odor annoyance depending on the health status.

Individual values of the odds ratios also allow stating that the high intensity of odors from the sludge dryer as well as from the primary settling tanks increases the chance of experiencing odor nuisance by over two times (Table 5). Also, for people who are focused on action in a stressful situation (the occurrence of smell) and those who describe their health as "bad", this chance increases twice. More prolonged exposure to odors (low intensity) minimizes the chance of experiencing the nuisance associated with the occurrence of odors.

Figure 18 presents graphs of the predicted probability of occurrence of odor annoyance depending on the intensity of odor for three different frequencies.

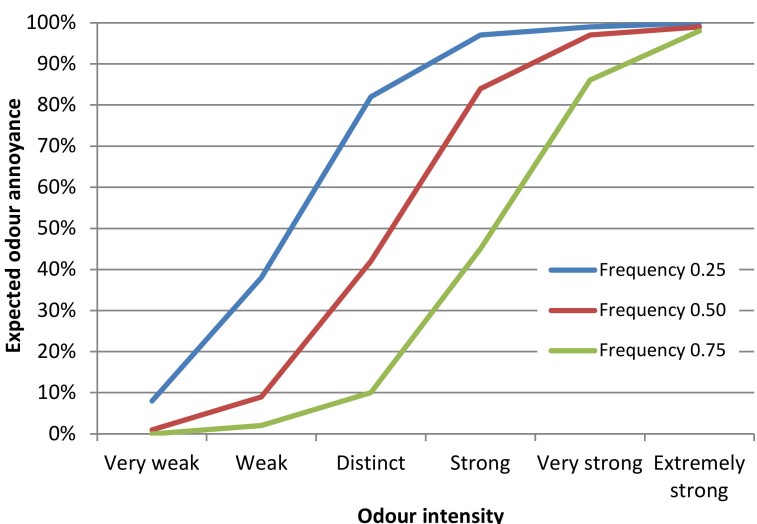

**Figure 18.** Dependence of the probability of occurrence of odor annoyance depending on the intensity of the smell.

The probability of occurrence of odor nuisance increases the fastest, at the lowest odor intensity values, in the case of the lowest frequency, and the slowest at much higher intensity values with the almost constant occurrence of odor (Figure 18). This may be because extended exposure to the fragrance causes people to become accustomed to it over time and in some cases, even causes a complete lack of the perception of a given fragrance [29].

Figure 19 shows the dependence of the probability of occurrence of odor nuisance depending on the frequency of odor, for three different intensities. In the case of a low odor intensity, increasing its incidence causes a decrease in perceived nuisance to 0, while in the case of maximum intensity of 6, even a high frequency of odor reduces only slightly the perceived annoyance. This may be since, although at high frequencies of smell, residents get used to it and do not feel it, at low intensity, its very presence with such a high intensity can cause their irritability.

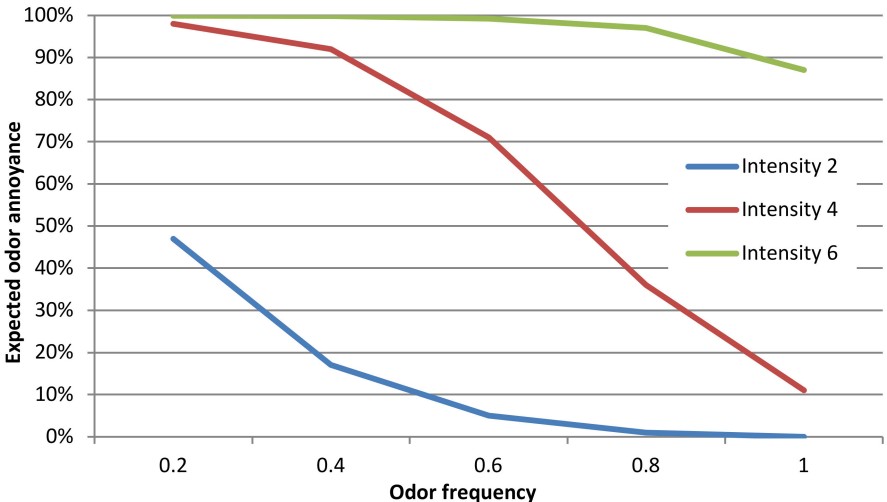

**Figure 19.** Dependence of the probability of occurrence of odor annoyance depending on the frequency of smell.

The HED variable for the hedonic tone of the fragrance did not enter the model. However, in the model, there were variables DRYER, SEDIMENT, and SETTLERS. They were describing the intensities of types of odors, which were assessed as unpleasant. Only the smell from bioreactors was determined at level 6 (on the primary scale +1, i.e., positive), which would confirm the results of earlier studies [27,30,31], indicating a clear impact of neutral and unpleasant odors on the irritation of residents. Also, contrary to expectations arising from previous studies [30], the model did not include such variables as age, gender, and education.

It cannot be 100% ruled out that other factors, which were not considered, influence the perception of odor annoyance in the residents of the housing estate located near the sewage treatment plant under investigation. However, the analysis presented above allows concluding that when the smell occurs irregularly, but with high intensity, the nonolfactory factors, such as parallel environmental pollution or a well-targeted response to stress, play a significant role in this process.

The analysis of the ROC curve results, which is a measure of model fit goodness, determining its good predictive properties, showed that the obtained model can be used as a predictor (AUC field = 0.82) in the assessment of potential perception of odor annoyance in areas located near the source of odor emission.

## 5. Conclusions

It has been proven in the work that the perception of odor annoyance is influenced by both the occurrence of smell in a given area (frequency and intensity) as well as other external, nonolfactory

factors, such as parallel environmental pollution (noise), the perceived state of health, or the degree of social activity.

A reliable mathematical model has been created to assess the source of perceived odor nuisance (the H-L test $p = 0.1134$, $R_N^2 = 0.53$, and AUC field = 0.82). Its results testify to a good fit of the developed model to the input data. This model can also be used as a predictor in assessing the potential perceived odor nuisance in areas located near the source of the odor emission.

The impact of nonolfactory factors on the perception of odor annoyance makes the following essential:

- At the investment planning stage, do not accumulate many facilities that could potentially have a negative impact on the environment and people (e.g., city beltway, sewage treatment plant, and chemical industry plant),
- Inform residents about the harmfulness of the smell or lack of it, to minimize the feeling of poor health,
- Treat complaints about the odor nuisance of inhabitants of areas exposed to the smell seriously as an indicator of the current state of the environment,
- Carry out eco-mediation and social consultations in which all interested parties must cooperate—authorities, residents' representatives, as well as owners of the facility potentially causing odor emissions.

The subject of further research in this regard should be the extension of the resulting model, considering the sources of odor emissions of various origins and the broadest possible group of respondents. This would allow for additional factors to be included in the model, which may also affect the degree of odor nuisance. It would be important to create a good model for the odor's dispersion and to assess the odor impact in the field, which could be input to the model, together with the possibility of obtaining statistical data. This would allow using the model developed in this work to forecast the perceived odor annoyance, as well as limiting the size of field measurements while maintaining high-quality results.

**Author Contributions:** Conceptualization, A.W. and J.Z.; methodology, A.W.; software, A.W.; validation, A.W. and J.Z.; formal analysis, A.W.; investigation, A.W.; resources, A.W. and J.Z.; data curation, A.W.; writing—original draft preparation, A.W.; writing—review and editing, A.W. and J.Z.; visualization, A.W.; supervision, J.Z.; project administration, A.W. and J.Z.; funding acquisition, J.Z. All authors have read and agreed to the published version of the manuscript.

**Funding:** The research was financed from the funds of order No. 600392 carried out at the Institute of Environmental Protection Engineering of the Wroclaw University of Technology. The APC was co-financed within the "Excellent Science" program of the Polish Ministry of Science and Higher Education.

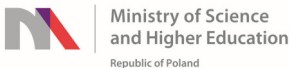

**Conflicts of Interest:** The authors declare no conflict of interest.

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
