# Peer review of "Odor Annoyance Assessment by Using Logistic Regression on an Example of the Municipal Sector"

_sustainability, doi:10.3390/su12156102_

Round 1

Reviewer 1 Report

The article is very interesting and points out the aspect of significance of olfactometric field tests. It is a very important issue, especially since, as the authors stressed, typical - albeit, unfortunately, commonly used - models do not include, among others, changes in weather conditions, terrain, or variable emission of pollutants. As also authors wrote, the model aren't very reliable as determining the area exposed to the effects of odours in the plant. So, field reconnaissances and field reseach are extremely important in such studies. The surveys used by the authors are also very important. Therefore, the authors have taken the right and insightful approach.

Nevertheless, I have some detailed questions and comments.

Lines 119-121 The authors wrote that "for this purpose, the Weber-Fechner factor Fw was used. Based on earlier studies, in which coefficients between 1.5 and 2.5 were tested, it was found that the coefficient 2 was best matched". So this is the authors' assumption based on literature studies? The authors did not try to determine the Weber-Fechner factor for this particular treatment plant? Alternatively, for specific sources of this plant (because such a complex facility is difficult to treat as a whole)?

Lines 219-220 Figure 3. The description is quite far ahead of this drawing (lines 187-190), which is understandable due to the layout of the text and pictures, but perhaps it would be reasonable to indicate in the description of the drawing that a rectangle marks the area of the plant?

Lines 222-224 Figure 4. I do not see any reason to introduce an additional dimension to the pie chart. Why is this chart presented as 3D, which, in my opinion, disturbs the reception in almost every case?

Figures 5-10. Could the authors consider using white (not transparent) background for the legend? Then the effect of overlapping the y-axis auxiliary lines on the legend would be avoided.

In the discussion section, there is no reference to the results of other researchers. Yes, the authors wrote (lines 362-365) that "...which would confirm the results of earlier studies [14,16-17] indicating a clear impact of neutral and unpleasant odours on the irritation of residents. Also, contrary to expectations arising from previous studies [16], the model did not include such variables as age, gender and education", but in my opinion this discussion is too poor. Could the authors extend it by referring to more specific results of other researchers?

Author Response

Dear Reviewer,

Thank you very much for all your comments and questions. Please, find the answers below.

  1. Lines 119-121 The authors wrote that "for this purpose, the Weber-Fechner factor Fw was used. Based on earlier studies, in which coefficients between 1.5 and 2.5 were tested, it was found that the coefficient 2 was best matched". So this is the authors' assumption based on literature studies? The authors did not try to determine the Weber-Fechner factor for this particular treatment plant? Alternatively, for specific sources of this plant (because such a complex facility is difficult to treat as a whole)?

Authors response:

That is correct. Authors did not determine the Weber-Fechner factor, neither for the whole plant, nor for any specific sources of the plant. We were focused on creating the model and the olfactometry part of the project was quite separate. What is even more important, the coefficient 2 was taken not only based on ‘some literature’. This decision was made primarily based on many discussion with one of this literature author’s (Kirsten Sucker) during 6 months training in Germany, taking into account the profile of the plant. Also, this studies was prepared for the Ministry of Environment in Germany and was a huge project, which analysed many aspects of this field. Nevertheless, it would be advisable to determine the coefficient for further studies in the future.

  1. Lines 219-220 Figure 3. The description is quite far ahead of this drawing (lines 187-190), which is understandable due to the layout of the text and pictures, but perhaps it would be reasonable to indicate in the description of the drawing that a rectangle marks the area of the plant?

Authors response:

The description (fig 3) is quite far from the drawing indeed. Unfortunately, it was very hard to place it somewhere closer and still maintain an appropriate structure of the paper. That is why, and we did this according to instructions for authors, we placed the page number next to it, so thank you for understanding.

It is, however, a very good suggestion to add information about the meaning of the rectangle on this figure. Please, find attached revised paper with changes.

  1. Lines 222-224 Figure 4. I do not see any reason to introduce an additional dimension to the pie chart. Why is this chart presented as 3D, which, in my opinion, disturbs the reception in almost every case?

Authors response:

To be honest, we have not noticed any inconveniences due to this fact before, but there are of course no obstacles to change it. Especially if it caused discomfort to the recipient. Please, find attached revised paper.

  1. Figures 5-10. Could the authors consider using white (not transparent) background for the legend? Then the effect of overlapping the y-axis auxiliary lines on the legend would be avoided.

Authors response:

Thank you very for this comment. It actually caused problems and somehow we did not get the idea to simply change the background of the legend. It was now done, so please find attached the revised paper.

  1. In the discussion section, there is no reference to the results of other researchers. Yes, the authors wrote (lines 362-365) that "...which would confirm the results of earlier studies [14,16-17] indicating a clear impact of neutral and unpleasant odours on the irritation of residents. Also, contrary to expectations arising from previous studies [16], the model did not include such variables as age, gender and education", but in my opinion this discussion is too poor. Could the authors extend it by referring to more specific results of other researchers?

Authors response:

We expanded the paper with additional references (please find attached revised work).

Also, we agree that the number of literature references is not impressive. However, the same can be said about the number of researcher doing studies in this narrow field. There are of course researches concerning filed measurements and odour nuisance surveys, but separate. With our approach, it is hardly possible to find a paper we could use to compare our results.

We truly hope that the supplement we did is now good enough.

Reviewer 2 Report

The presented manuscript shows the assessment of odour annoyance using logistic regression on an example of the municipal sector. In general, the work is prepared correctly, the methods are sufficiently described, and the results are interesting. However, the reviewer asks you to respond to the following comments:

  1. Title - in my opinion the form "logistic regresision" is more correct than "regression logistic"
  2. What the measurements carried out using trained group of testers looked like? Did they breathe odorless air before taking the measurement? If not, then has the phenomenon of olfactory adaptation not affected here? Please comment.
  3. Table 3: The average values were rounded? In the case of average measurement values, they should rather not be integer values.
  4. Line 380: "A reliable mathematical model has been created to assess the source of perceived odour nuisance. Its results testify to a good fit of the developed model to the input data." The "good fit" of the model was proved using a validation parameter RN2 - please define it. Entering the value itself is of course correct, but showing the expected/observed plot would be advisable.
  5. Please add references to surveys and field studies related to the occurrence of odour nuisances. 17 references are definitely not enough.
  6. Please remove borders in Figures.

Author Response

Dear Reviewer,

Thank you very much for all your comments and questions. Please, find the answers below.

  1. Title - in my opinion the form "logistic regresision" is more correct than "regression logistic"

Authors response:

Thank you for the correction, of course “logistic regression” is more correct.

  1. What the measurements carried out using trained group of testers looked like? Did they breathe odorless air before taking the measurement? If not, then has the phenomenon of olfactory adaptation not affected here? Please comment.

Authors response:

Experts did not breathe odourless air before taking the measurements. The phenomenon of odour adaptation is not a problem here, however. The testers knew the previously selected smells and were able to identify them and extract them from the surrounding air. Field measurements were carried out according to VDI 3940 Part .3, Measurement of odour in ambient air by field inspections - Determination of odour intensity and hedonic odour tone, Verein DeutscherIngenieure, Berlin, Beuth Verlag, 2010. Moreover, “testers” was a trained group of adults who were first and on a regular base tested on n-Butanol in the laboratory and who fulfilled the criteria set out in the standard “Polish standard PN-EN 13725:2007: Air quality - Determination of odour concentration by dynamic olfactometry. Polish Committee for Standardization” (honestly, our big mistake not to put the standard as a reference). A single measurement last 10 minutes, so above all, this period would not be long enough to adapt to the odour. What is also important, during single measurements, the experts would smell the odour in very different moments, sometimes only at the end, other time just in the middle of the measurement period.

  1. Table 3: The average values were rounded? In the case of average measurement values, they should rather not be integer values.

Authors response:

The scale for both intensity and hedonic tone consists of integer numbers only. We decided that the presentation of the results in this way would not be incorrect, and at the same time more legible.

  1. Line 380: "A reliable mathematical model has been created to assess the source of perceived odour nuisance. Its results testify to a good fit of the developed model to the input data." The "good fit" of the model was proved using a validation parameter RN- please define it. Entering the value itself is of course correct, but showing the expected/observed plot would be advisable.

Authors response:

We supplemented the missing data in the summary. Additionally, we developed a part of the methodology and results with the previously performed ROC analysis. We hope there are no more doubts now. Please find attached revised paper.

  1. Please add references to surveys and field studies related to the occurrence of odour nuisances. 17 references are definitely not enough.

Authors response:

We expanded the paper with additional references (please find attached revised work).

Also, we agree that the number of literature references is not impressive. However, the same can be said about the number of researcher doing studies in this narrow field. There are of course researches concerning filed measurements and odour nuisance surveys, but separate. With our approach, it is hardly possible to find a paper we could use to compare our results.

We truly hope that the supplement we did is now good enough.

  1. Please remove borders in Figures.

Authors response:

It has been done. Please, see attached revised version.

Reviewer 3 Report

I think it is a quite relevant field and it should be published.

Author Response

Dear Reviewer,

Thank you very much for all your comments and questions. Please, find the answers below.

Authors response:

Thank you for the comment. The work has been a bit revised anyway, so please find attached a new version.
